# Sign-OPT: A Query-Efficient Hard-label Adversarial Attack

**Minhao Cheng**[1*], **Simranjit Singh**[1*], **Patrick Chen**[1], **Pin-Yu Chen**[2], **Sijia Liu**[2], **Cho-Jui Hsieh**[1]
[1]Department of Computer Science, UCLA, [2]IBM Research
{mhcheng, simranjit, patrickchen, chohsieh}@cs.ucla.edu
{pin-yu.chen, sijia.liu}@ibm.com

## Abstract

We study the most practical problem setup for evaluating adversarial robustness of a machine learning system with limited access: the hard-label black-box attack setting for generating adversarial examples, where limited model queries are allowed and only the decision is provided to a queried data input. Several algorithms have been proposed for this problem but they typically require huge amount (>20,000) of queries for attacking one example. Among them, one of the state-of-the-art approaches (Cheng et al., 2019) showed that hard-label attack can be modeled as an optimization problem where the objective function can be evaluated by binary search with additional model queries, thereby a zeroth order optimization algorithm can be applied. In this paper, we adopt the same optimization formulation but propose to directly estimate the sign of gradient at any direction instead of the gradient itself, which enjoys the benefit of single query. Using this single query oracle for retrieving sign of directional derivative, we develop a novel query-efficient Sign-OPT approach for hard-label black-box attack. We provide a convergence analysis of the new algorithm and conduct experiments on several models on MNIST, CIFAR-10 and ImageNet. We find that Sign-OPT attack consistently requires $5\times$ to $10\times$ fewer queries when compared to the current state-of-the-art approaches, and usually converges to an adversarial example with smaller perturbation.

## 1 Introduction

It has been shown that neural networks are vulnerable to adversarial examples (Szegedy et al., 2016; Goodfellow et al., 2015; Carlini & Wagner, 2017; Athalye et al., 2018). Given a victim neural network model and a correctly classified example, an adversarial attack aims to compute a small perturbation such that with this perturbation added, the original example will be misclassified. Many adversarial attacks have been proposed in the literature. Most of them consider the white-box setting, where the attacker has full knowledge about the victim model, and thus gradient based optimization can be used for attack. Popular Examples include C&W (Carlini & Wagner, 2017) and PGD (Madry et al., 2017) attacks. On the other hand, some more recent attacks have considered the probability black-box setting where the attacker does not know the victim model's structure and weights, but can iteratively query the model and get the corresponding probability output. In this setting, although gradient (of output probability to the input layer) is not computable, it can still be estimated using finite differences, and algorithms many attacks are based on this (Chen et al., 2017; Ilyas et al., 2018a; Tu et al., 2019; Jun et al., 2018).

In this paper, we consider the most challenging and practical attack setting – hard-label black-box setting – where the model is hidden to the attacker and the attacker can only make queries and get the corresponding hard-label decisions (e.g., predicted labels) of the model. A commonly used algorithm proposed in this setting, also called Boundary attack (Brendel et al., 2017), is based on random walks on the decision surface, but it does not have any convergence guarantee. More recently, Cheng et al. (2019) showed that finding the minimum adversarial perturbation in the hard-label setting can be reformulated as another optimization problem (we call this Cheng's formulation in this paper). This

---

*Equal Contribution.

new formulation enjoys the benefit of having a smooth boundary in most tasks and the function value is computable using hard-label queries. Therefore, the authors of (Cheng et al., 2019) are able to use standard zeroth order optimization to solve the new formulation. Although their algorithm converges quickly, it still requires large number of queries (e.g., 20,000) for attacking a single image since every function evaluation of Cheng's formulation has to be computed using binary search requiring tens of queries.

In this paper, we follow the same optimization formulation of (Cheng et al., 2019) which has the advantage of smoothness, but instead of using finite differences to estimate the magnitude of directional derivative, we propose to evaluate its sign using only **a single query**. With this single-query sign oracle, we design novel algorithms for solving the Cheng's formulation, and we theoretically prove and empirically demonstrate the significant reduction in the number of queries required for hard-label black box attack.

Our contribution are summarized below:

- **Novelty in terms of adversarial attack.** We elucidate an efficient approach to compute the sign of directional derivative of Cheng's formulation using a single query, and based on this technique we develop a novel optimization algorithm called Sign-OPT for hard-label black-box attack.
- **Novelty in terms of optimization.** Our method can be viewed as a new zeroth order optimization algorithm that features fast convergence of signSGD. Instead of directly taking the sign of gradient estimation, our algorithm utilizes the scale of random direction. This make existing analysis inappropriate to our case, and we provide a new recipe to prove the convergence of this new optimizer.
- We conduct comprehensive experiments on several datasets and models. We show that the proposed algorithm consistently reduces the query count by 5–10 times across different models and datasets, suggesting a practical and query-efficient robustness evaluation tool. Furthermore, on most datasets our algorithm can find an adversarial example with smaller distortion compared with previous approaches.

## 2    RELATED WORK

**White-box attack**    Since it was firstly found that neural networks are easy to be fooled by adversarial examples (Goodfellow et al., 2015), a lot of work has been proposed in the white-box attack setting, where the classifier $f$ is completely exposed to the attacker. For neural networks, under this assumption, back-propagation can be conducted on the target model because both network structure and weights are known by the attacker. Algorithms including (Goodfellow et al., 2015; Kurakin et al., 2016; Carlini & Wagner, 2017; Chen et al., 2018; Madry et al., 2017) are then proposed based on gradient computation. Recently, the BPDA attack introduced by Athalye et al. (2018) bypasses some models with obfuscated gradients and is shown to successfully circumvent many defenses. In addition to typical attacks based on small $\ell_p$ norm perturbation, non-$\ell_p$ norm perturbations such as scaling or shifting have also been considered (Zhang et al., 2019).

**Black-box attack**    Recently, black-box setting is drawing rapidly increasing attention. In black-box setting, the attacker can query the model but has no (direct) access to any internal information inside the model. Although there are some works based on transfer attack (Papernot et al., 2017), we consider the query-based attack in the paper. Depending on the model's feedback for a given query, an attack can be classified as a soft-label or hard-label attack. In the soft-label setting, the model outputs a probability score for each decision. Chen et al. (2017) uses a finite difference in a coordinate-wise manner to approximately estimate the output probability changes and does a coordinate descent to conduct the attack. Ilyas et al. (2018a) uses Neural evolution strategy (NES) to approximately estimate the gradient directly. Later, some variants (Ilyas et al., 2018b; Tu et al., 2019) were proposed to utilize the side information to further speed up the attack procedure. Alzantot et al. (2019) uses a evolutionary algorithm as a black-box optimizer for the soft-label setting. Recently, Al-Dujaili & O'Reilly (2019) proposes SignHunter algorithm based on signSGD (Bernstein et al., 2018) to achieve faster convergence in the soft-label setting. The recent work (Al-Dujaili & O'Reilly, 2019) proposes SignHunter algorithm to achieve a more query-efficent sign estimate when crafting black-box adversarial examples through soft-label information.

In the hard-label case, only the final decision, i.e. the top-1 predicted class, is observed. As a result, the attacker can only make queries to acquire the corresponding hard-label decision instead of the probability outputs. Brendel et al. (2017) first studied this problem and proposed an algorithm based on random walks near the decision boundary. By selecting a random direction and projecting it onto a boundary sphere in each iteration, it aims to generate a high-quality adversarial example. Query-Limited attack (Ilyas et al., 2018a) tries to estimate the output probability scores with model query and turn the hard-label into a soft-label problem. Cheng et al. (2019) instead re-formalizes the hard-label attack into an optimization problem that finds a direction which could produce the shortest distance to decision boundary.

The recent arXiv paper (Chen et al., 2019) applied the zeroth-order sign oracle to improve Boundary attack, and also demonstrated significant improvement. The major differences to our algorithm are that we propose a new zeroth-order gradient descent algorithm, provide its algorithmic convergence guarantees, and aim to improve the query complexity of the attack formulation proposed in (Cheng et al., 2019). For completeness, we also compare with this method in Section A.1. Moreover, (Chen et al., 2019) uses one-point gradient estimate, which is unbiased but may encounter larger variance compared with the gradient estimate in our paper. Thus, we can observe in Section A.1 that although they are slightly faster in the initial stage, Sign-OPT will catch up and eventually lead to a slightly better solution.

## 3 PROPOSED METHOD

We follow the same formulation in (Cheng et al., 2019) and consider the hard-label attack as the problem of finding the direction with shortest distance to the decision boundary. Specifically, for a given example $x_0$, true label $y_0$ and the hard-label black-box function $f : \mathbb{R}^d \to \{1, \ldots, K\}$, the objective function $g : \mathbb{R}^d \to \mathbb{R}$ (for the untargeted attack) can be written as:

$$\min_{\boldsymbol{\theta}} g(\boldsymbol{\theta}) \text{ where } g(\boldsymbol{\theta}) = \arg\min_{\lambda > 0} \left( f(x_0 + \lambda \frac{\boldsymbol{\theta}}{\|\boldsymbol{\theta}\|}) \neq y_0 \right). \tag{1}$$

It has been shown that this objective function is usually smooth and the objective function $g$ can be evaluated by a binary search procedure locally. At each binary search step, we query the function $f(\boldsymbol{x}_0 + \lambda \frac{\boldsymbol{\theta}}{\|\boldsymbol{\theta}\|})$ and determine whether the distance to decision boundary in the direction $\boldsymbol{\theta}$ is greater or smaller than $\lambda$ based on the hard-label prediction[1].

As the objective function is computable, the directional derivative of $g$ can be estimated by finite differences:

$$\hat{\nabla}g(\boldsymbol{\theta}; \mathbf{u}) := \frac{g(\boldsymbol{\theta} + \epsilon\mathbf{u}) - g(\boldsymbol{\theta})}{\epsilon} \mathbf{u} \tag{2}$$

where $\boldsymbol{u}$ is a random Gaussian vector and $\epsilon > 0$ is a very small smoothing parameter. This is a standard zeroth order oracle for estimating directional derivative and based on this we can apply many different zeroth order optimization algorithms to minimize $g$.

For example, Cheng et al. (2019) used the Random Derivative Free algorithm Nesterov & Spokoiny (2017) to solve problem (1). However, each computation of (2) requires many hard-label queries due to binary search, so Cheng et al. (2019) still requires a huge number of queries despite having fast convergence.

In this work, we introduce an algorithm that hugely improves the query complexity over Cheng et al. (2019). Our algorithm is based on the following key ideas: (i) one does not need very accurate values of directional derivative in order to make the algorithm converge, and (ii) there exists an **imperfect but informative estimation** of directional derivative of $g$ that can be computed by a single query.

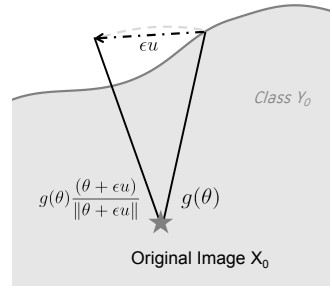

Figure 1: Illustration

---

[1]Note that binary search only works in a small local region; in more general case $g(\boldsymbol{\theta})$ has to be computed by a fine-grained search plus binary search, as discussed in Cheng et al. (2019).

---

**Algorithm 1:** Sign-OPT attack

---

**Input**: Hard-label model $f$, original image $x_0$, initial $\boldsymbol{\theta}_0$ ;
**for** $t = 1, 2, \ldots, T$ **do**

    Randomly sample $u_1, \ldots, u_Q$ from a Gaussian or Uniform distribution;

    Compute $\hat{\boldsymbol{g}} \leftarrow \frac{1}{Q} \sum_{q=1}^{Q} \text{sign}(g(\boldsymbol{\theta}_t + \epsilon \mathbf{u}_q) - g(\boldsymbol{\theta}_t)) \cdot \mathbf{u}_q$ ;

    Update $\boldsymbol{\theta}_{t+1} \leftarrow \boldsymbol{\theta}_t - \eta \hat{\boldsymbol{g}}$ ;

    Evaluate $g(\boldsymbol{\theta}_t)$ using the same search algorithm in Cheng et al. (2019) ;

**end**

---

### 3.1 A SINGLE QUERY ORACLE

As mentioned before, the previous approach requires computing $g(\boldsymbol{\theta} + \epsilon\boldsymbol{u}) - g(\boldsymbol{\theta})$ which consumes a lot of queries. However, based on the definition of $g(\cdot)$, we can compute the sign of this value $\text{sign}(g(\boldsymbol{\theta} + \epsilon\boldsymbol{u}) - g(\boldsymbol{\theta}))$ using a single query. Considering the untargeted attack case, the sign can be computed by

$$\text{sign}(g(\boldsymbol{\theta} + \epsilon\mathbf{u}) - g(\boldsymbol{\theta})) = \begin{cases} +1, & f(x_0 + g(\boldsymbol{\theta})\frac{(\boldsymbol{\theta}+\epsilon\mathbf{u})}{\|\boldsymbol{\theta}+\epsilon\mathbf{u}\|}) = y_0, \\ -1, & \text{Otherwise.} \end{cases} \tag{3}$$

This is illustrated in Figure 1. Essentially, for a new direction $\boldsymbol{\theta} + \epsilon\mathbf{u}$, we test whether a point at the original distance $g(\boldsymbol{\theta})$ from $x_0$ in this direction lies inside or outside the decision boundary, i.e. if the produced perturbation will result in a wrong prediction by classifier. If the produced perturbation is outside the boundary i.e. $f(x_0 + g(\boldsymbol{\theta})\frac{(\boldsymbol{\theta}+\epsilon\mathbf{u})}{\|\boldsymbol{\theta}+\epsilon\mathbf{u}\|}) \neq y_0$, the new direction has a smaller distance to decision boundary, and thus giving a smaller value of $g$. It indicates that $\mathbf{u}$ is a descent direction to minimize $g$.

### 3.2 SIGN-OPT ATTACK

By sampling random Gaussian vector $Q$ times, we can estimate the imperfect gradient by

$$\hat{\nabla} g(\boldsymbol{\theta}) \approx \hat{\boldsymbol{g}} := \sum_{q=1}^{Q} \text{sign}(g(\boldsymbol{\theta} + \epsilon\mathbf{u}_q) - g(\boldsymbol{\theta}))\mathbf{u}_q, \tag{4}$$

which only requires $Q$ queries. We then use this imperfect gradient estimate to update our search direction $\boldsymbol{\theta}$ as $\boldsymbol{\theta} \leftarrow \boldsymbol{\theta} - \eta\hat{\mathbf{g}}$ with a step size $\eta$ and use the same search procedure to compute $g(\boldsymbol{\theta})$ up to a certain accuracy. The detailed procedure is shown in Algorithm 1.

We note that Liu et al. (2019) designed a Zeroth Order SignSGD algorithm for soft-label black box attack (not hard-label setting). They use $\hat{\nabla} g(\boldsymbol{\theta}) \approx \hat{\boldsymbol{g}} := \sum_{q=1}^{Q} \text{sign}(g(\boldsymbol{\theta} + \epsilon\mathbf{u}_q) - g(\boldsymbol{\theta})\mathbf{u}_q)$ and shows that it could achieve a comparable or even better convergence rate than zeroth order stochastic gradient descent by using only sign information of gradient estimation. Although it is possible to combine ZO-SignSGD with our proposed single query oracle for solving hard-label attack, their estimator will take sign of the whole vector and thus ignore the direction of $\mathbf{u}_q$, which leads to slower convergence in practice (please refer to Section 4.4 and Figure 5(b) for more details).

To the best of our knowledge, no previous analysis can be used to prove convergence of Algorithm 1. In the following, we show that Algorithm 1 can in fact converge and furthermore, with similar convergence rate compared with (Liu et al., 2019) despite using a different gradient estimator.

**Assumption 1.** *Function $g(\theta)$ is L-smooth with a finite value of L.*

**Assumption 2.** *At any iteration step t, the gradient of the function g is upper bounded by $\|\nabla g(\boldsymbol{\theta}_t)\|_2 \leq \sigma$.*

**Theorem 3.1.** *Suppose that the conditions in the assumptions hold, and the distribution of gradient noise is unimodal and symmetric. Then, Sign-OPT attack with learning rate $\eta_t = O(\frac{1}{Q\sqrt{dT}})$ and $\epsilon = O(\frac{1}{dT})$ will give following bound on $\mathbb{E}[\|\nabla g(\boldsymbol{\theta})\|_2]$:*

$$\mathbb{E}[\|\nabla g(\boldsymbol{\theta})\|_2] = O(\frac{\sqrt{d}}{\sqrt{T}} + \frac{d}{\sqrt{Q}}).$$

The proof can be found in subsection A.2. The main difference with the original analysis provided by Liu et al. (2019) is that they only only deal with sign of each element, while our analysis also takes the magnitudes of each element of $\boldsymbol{u}_q$ into account.

### 3.3 Other gradient estimations

Note that the value $\text{sign}(g(\boldsymbol{\theta} + \epsilon \boldsymbol{u}) - g(\boldsymbol{\theta}))$ computed by our single query oracle is actually the sign of the directional derivative:

$$\text{sign}(\langle \nabla g(\boldsymbol{\theta}), \boldsymbol{u} \rangle) = \text{sign}(\lim_{\epsilon \to \infty} \frac{g(\boldsymbol{\theta} + \epsilon \boldsymbol{u}) - g(\boldsymbol{\theta})}{\epsilon}) = \text{sign}(g(\boldsymbol{\theta} + \epsilon \boldsymbol{u}) - g(\boldsymbol{\theta})) \text{ for a small } \epsilon.$$

Therefore, we can use this information to estimate the original gradient. The Sign-OPT approach in the previous section uses $\sum_q \text{sign}(\langle \nabla g(\boldsymbol{\theta}), \boldsymbol{u}_q \rangle) \boldsymbol{u}_q$ as an estimation of gradient. Let $y_q := \text{sign}(\langle \nabla g(\boldsymbol{\theta}), \boldsymbol{u}_q \rangle)$, a more accurate gradient estimation can be cast as the following constraint optimization problem:

$$\text{Find a vector } \boldsymbol{z} \text{ such that } \text{sign}(\langle \boldsymbol{z}, \boldsymbol{u}_q \rangle) = y_q \;\; \forall q = 1, \dots, Q.$$

Therefore, this is equivalent to a hard constraint SVM problem where each $\boldsymbol{u}_q$ is a training sample and $y_q$ is the corresponding label. The gradient can then be recovered by solving the following quadratic programming problem:

$$\min_{\boldsymbol{z}} \; \boldsymbol{z}^T \boldsymbol{z} \;\; \text{s.t.} \;\; \boldsymbol{z}^T \boldsymbol{u}_q \geq y_q, \;\; \forall q = 1, \dots, Q. \tag{5}$$

By solving this problem, we can get a good estimation of the gradient. As explained earlier, each $y_q$ can be determined with a single query. Therefore, we propose a variant of Sign-OPT, which is called SVM-OPT attack. The detailed procedure is shown in Algorithm 2. We will present an empirical comparison of our two algorithms in subsection 4.1.

---

**Algorithm 2:** SVM-OPT attack

---

**Input**: Hard-label model $f$, original image $\boldsymbol{x}_0$, initial $\boldsymbol{\theta}_0$ ;
**for** $t = 1, 2, \dots, T$ **do**
    Sample $\boldsymbol{u}_1, \dots, \boldsymbol{u}_Q$ from Gaussian or orthogonal basis ;
    Solve $\boldsymbol{z}$ defined by (5) ;
    Update $\boldsymbol{\theta}_{t+1} \leftarrow \boldsymbol{\theta}_t - \eta \boldsymbol{z}$ ;
    Evaluate $g(\boldsymbol{\theta}_t)$ using search algorithm in (Cheng et al., 2019) ;
**end**

---

## 4 Experimental Results

We evaluate the SIGN-OPT algorithm for attacking black-box models in a hard-label setting on three different standard datasets - MNIST (LeCun et al., 1998), CIFAR-10 (Krizhevsky et al.) and ImageNet-1000 (Deng et al., 2009) and compare it with existing methods. For fair and easy comparison, we use the CNN networks provided by (Carlini & Wagner, 2017), which have also been used by other previous hard-label attacks as well. Specifically, for both MNIST and CIFAR-10, the model consists of nine layers in total - four convolutional layers, two max-pooling layers and two fully-connected layers. Further details about implementation, training and parameters are available on (Carlini & Wagner, 2017). As reported in (Carlini & Wagner, 2017) and (Cheng et al., 2019), we were able to achieve an accuracy of 99.5% on MNIST and 82.5% on CIFAR-10. We use the pretrained Resnet-50 (He et al., 2016) network provided by torchvision (Marcel & Rodriguez, 2010) for ImageNet-1000, which achieves a Top-1 accuracy of 76.15%.

In our experiments, we found that Sign-OPT and SVM-OPT perform quite similarly in terms of query efficiency. Hence we compare only Sign-OPT attack with previous approaches and provide a comparison between Sign-OPT and SVM-OPT in subsection 4.1. We compare the following attacks:

- **Sign-OPT attack** (black box): The approach presented in this paper.
- **Opt-based attack** (black box): The method proposed in Cheng et al. (2019) where they use Randomized Gradient-Free method to optimize the same objective function. We use the implementation provided at `https://github.com/LeMinhThong/blackbox-attack`.

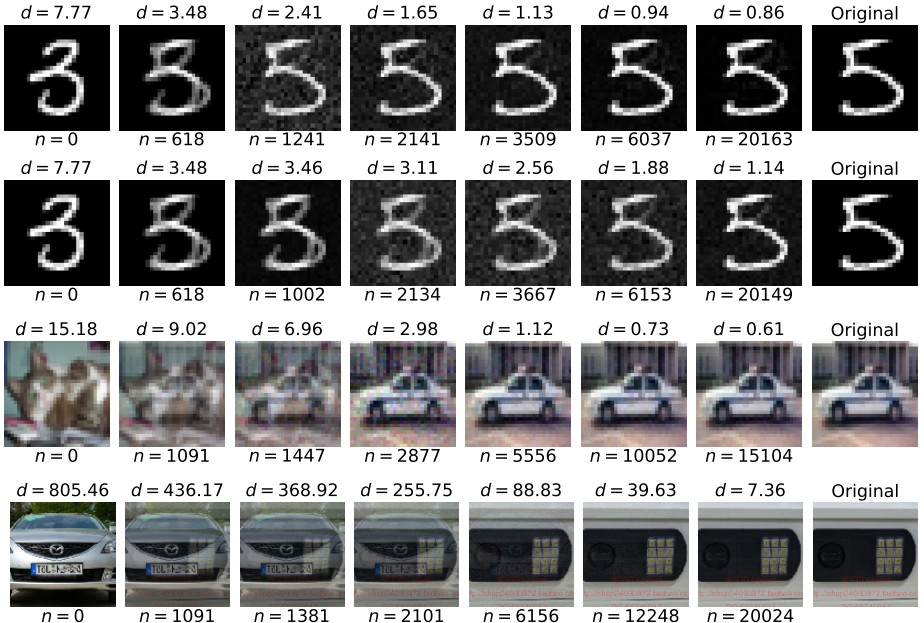

Figure 2: Example of Sign-OPT targeted attack. $L_2$ distortions and queries used are shown above and below the images. First two rows: Example comparison of Sign-OPT attack and OPT attack. Third and fourth rows: Examples of Sign-OPT attack on CIFAR-10 and ImageNet

- **Boundary attack** (black box): The method proposed in Brendel et al. (2017). This is compared only in $L_2$ setting as it is designed for the same. We use the implementation provided in Foolbox (`https://github.com/bethgelab/foolbox`).

- **Guessing Smart Attack** (black box): The method proposed in (Brunner et al., 2018). This attack enhances boundary attack by biasing sampling towards three priors. Note that one of the priors assumes access to a similar model as the target model and for a fair comparison we do not incorporate this bias in our experiments. We use the implementation provided at `https://github.com/ttbrunner/biased_boundary_attack`.

- **C&W attack** (white box): One of the most popular methods in the white-box setting proposed in Carlini & Wagner (2017). We use C&W $L_2$ norm attack as a baseline for the white-box attack performance.

For each attack, we randomly sample 100 examples from validation set and generate adversarial perturbations for them. For untargeted attack, we only consider examples that are correctly predicted by model and for targeted attack, we consider examples that are already not predicted as target label by the model. To compare different methods, we mainly use *median distortion* as the metric. Median distortion for $x$ queries is the median adversarial perturbation of all examples achieved by a method using less than $x$ queries. Since all the hard-label attack algorithms will start from an adversarial exmample and keep reduce the distortion, if we stop at any time they will always give an adversarial example and medium distortion will be the most suitable metric to compare their performance. Besides, we also show *success rate (SR)* for $x$ queries for a given threshold ($\epsilon$), which is the percentage of number of examples that have achieved an adversarial perturbation below $\epsilon$ with less than $x$ queries. We evaluate success rate on different thresholds which depend on the dataset being used. For comparison of different algorithms in each setting, we chose the same set of examples across all attacks.

**Implementation details**: To optimize algorithm 1, we estimate the step size $\eta$ using the same line search procedure implemented in Cheng et al. (2019). At the cost of a relatively small number of queries, this provides significant speedup in the optimization. Similar to Cheng et al. (2019), $g(\theta)$ in last step of algorithm 1 is approximated via binary search. The initial $\theta_0$ in algorithm 1 is calculated

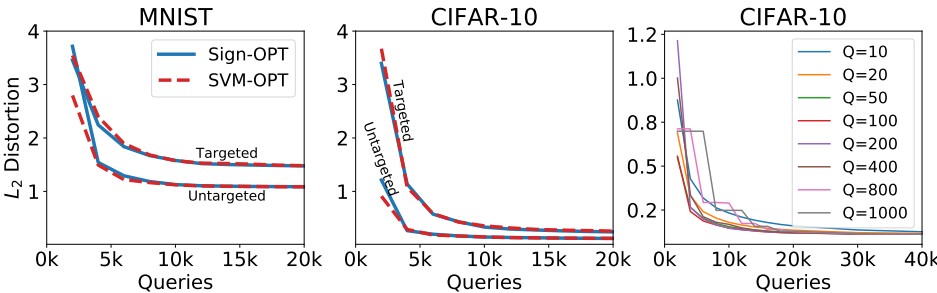

Figure 3: Median $L_2$ distortion vs Queries. First two: Comparison of Sign-OPT and SVM-OPT attack for MNIST and CIFAR-10. Third: Performance of Sign-OPT for different values of $Q$.

by evaluating $g(\theta)$ on 100 random directions and taking the best one. We provide our implementation publicly[2].

## 4.1 COMPARISON BETWEEN SIGN-OPT AND SVM-OPT

In our experiments, we found that the performance in terms of queries of both these attacks is remarkably similar in all settings (both $L_2/L_\infty$ & Targeted/Untargeted) and datasets. We present a comparison for MNIST and CIFAR-10 ($L_2$ norm-based) for both targeted and untargeted attacks in Figure 3. We see that the median distortion achieved for a given number of queries is quite on part for both Sign-OPT and SVM-OPT.

**Number of queries per gradient estimate**: In Figure 3, we show the comparison of Sign-OPT attack with different values of $Q$. Our experiments suggest that $Q$ does not have an impact on the convergence point reached by the algorithm. Although, small values of $Q$ provide a noisy gradient estimate and hence delayed convergence to an adversarial perturbation. Large values of $Q$, on the other hand, require large amount of time per gradient estimate. After fine tuning on a small set of examples, we found that $Q = 200$ provides a good balance between the two. Hence, we set the value of $Q = 200$ for all our experiments in this section.

## 4.2 UNTARGETED ATTACK

In this attack, the objective is to generate an adversary from an original image for which the prediction by model is different from that of original image. Figure 4 provides an elaborate comparison of different attacks for $L_2$ case for the three datasets. Sign-OPT attack consistently outperforms the current approaches in terms of queries. Not only is Sign-OPT more efficient in terms of queries, in most cases it converges to a lower distortion than what is possible by other hard-label attacks. Furthermore, we observe Sign-OPT converges to a solution comparable with C&W white-box attack (better on CIFAR-10, worse on MNIST, comparable on ImageNet). This is significant for a hard-label attack algorithm since we are given very limited information.

We highlight some of the comparisons of Boundary attack, OPT-based attack and Sign-OPT attack ($L_2$ norm-based) in Table 1. Particularly for ImageNet dataset on ResNet-50 model, Sign-OPT attack reaches a median distortion below 3.0 in less than $30k$ queries while other attacks need more than $200k$ queries for the same.

## 4.3 TARGETED ATTACK

In targeted attack, the goal is to generate an adversarial perturbation for an image so that the prediction of resulting image is the same as a specified target. For each example, we randomly specify the target label, keeping it consistent across different attacks. We calculate the initial $\theta_0$ in algorithm 1 using 100 samples in target label class from training dataset and this $\theta_0$ is the same across different attacks. Figure 2 shows some examples of adversarial examples generated by Sign-OPT attack and the Opt-based attack. The first two rows show comparison of Sign-OPT and Opt attack respectively on an example from MNIST dataset. The figures show adversarial examples generated at almost

---

[2]https://github.com/cmhcbb/attackbox

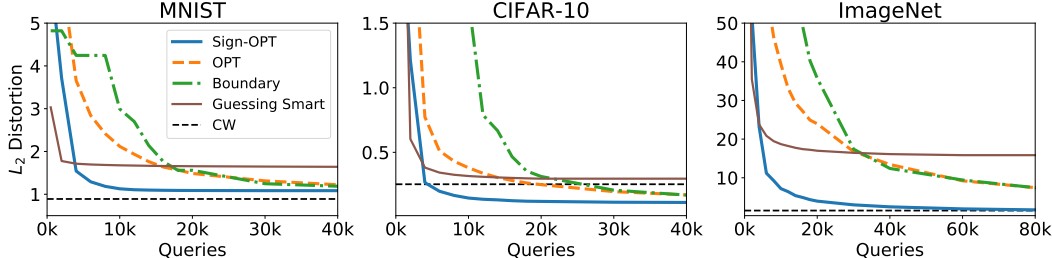

Figure 4: Untargeted attack: Median distortion vs Queries for different datasets.

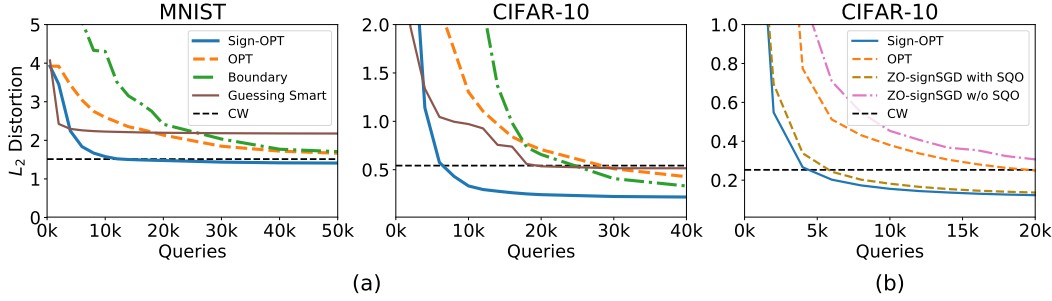

Figure 5: (a) Targeted Attack: Median distortion vs Queries of different attacks on MNIST and CIFAR-10. (b) Comparing Sign-OPT and ZO-SignSGD with and without single query oracle (SQO).

same number of queries for both attacks. Sign-OPT method generates an $L_2$ adversarial perturbation of 0.94 in $\sim 6k$ queries for this particular example while Opt-based attack requires $\sim 35k$ for the same. Figure 5 displays a comparison among different attacks in targeted setting. In our experiments, average distortion achieved by white box attack C&W for MNIST dataset is 1.51, for which Sign-OPT requires $\sim 12k$ queries while others need $> 120k$ queries. We present a comparison of success rate of different attacks for CIFAR-10 dataset in Figure 6 for both targeted and untargeted cases.

## 4.4 THE POWER OF SINGLE QUERY ORACLE

In this subsection, we conduct several experiments to prove the effectiveness of our proposed single query oracle in hard-label adversarial attack setting. ZO-SignSGD algorithm (Liu et al., 2019) is proposed for soft-label black box attack and we extend it into hard-label setting. A straightforward way is simply applying ZO-SignSGD to solve the hard-label objective proposed in Cheng et al. (2019), estimate the gradient using binary search as (Cheng et al., 2019) and take its sign. In Figure 5(b), we clearly observe that simply combining ZO-SignSGD and Cheng et al. (2019) is not efficient. With the proposed single query sign oracle, we can also reduce the query count of this method, as demonstrated in Figure 5(b). This verifies the effectiveness of single query oracle, which can universally improve many different optimization methods in the hard-label attack setting. To be noted, there is still improvement on Sign-OPT over ZO-SignSGD with single query oracle because instead

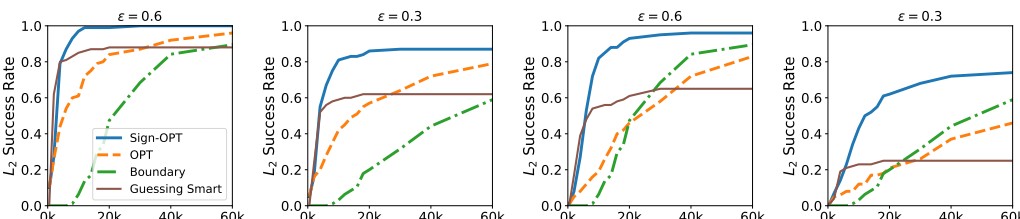

Figure 6: Success Rate vs Queries for CIFAR-10 ($L_2$ norm-based attack). First two and last two depict untargeted and targeted attacks respectively. Success rate threshold is at the top of each plot.

Table 1: $L_2$ Untargeted attack - Comparison of average $L_2$ distortion achieved using a given number of queries for different attacks. SR stands for success rate.

| | MNIST | | | CIFAR10 | | | ImageNet (ResNet-50) | | |
|---|---|---|---|---|---|---|---|---|---|
| | #Queries | Avg $L_2$ | SR($\epsilon = 1.5$) | #Queries | Avg $L_2$ | SR($\epsilon = 0.5$) | #Queries | Avg $L_2$ | SR($\epsilon = 3.0$) |
| Boundary attack | 4,000 | 4.24 | 1.0% | 4,000 | 3.12 | 2.3% | 4,000 | 209.63 | 0% |
| | 8,000 | 4.24 | 1.0% | 8,000 | 2.84 | 7.6% | 30,000 | 17.40 | 16.6% |
| | 14,000 | 2.13 | 16.3% | 12,000 | 0.78 | 29.2% | 160,000 | 4.62 | 41.6% |
| OPT attack | 4,000 | 3.65 | 3.0% | 4,000 | 0.77 | 37.0% | 4,000 | 83.85 | 2.0% |
| | 8,000 | 2.41 | 18.0% | 8,000 | 0.43 | 53.0% | 30,000 | 16.77 | 14.0% |
| | 14,000 | 1.76 | 36.0% | 12,000 | 0.33 | 61.0% | 160,000 | 4.27 | 34.0% |
| Guessing Smart | 4,000 | 1.74 | 41.0% | 4,000 | 0.29 | 75.0% | 4,000 | 16.69 | 12.0% |
| | 8,000 | 1.69 | 42.0% | 8,000 | 0.25 | 80.0% | 30,000 | 13.27 | 12.0% |
| | 14,000 | 1.68 | 43.0% | 12,000 | 0.24 | 80.0% | 160,000 | 12.88 | 12.0% |
| **Sign-OPT attack** | 4,000 | 1.54 | 46.0% | 4,000 | 0.26 | 73.0% | 4,000 | 23.19 | 8.0% |
| | 8,000 | 1.18 | 84.0% | 8,000 | 0.16 | 90.0% | 30,000 | 2.99 | 50.0% |
| | 14,000 | 1.09 | 94.0% | 12,000 | 0.13 | 95.0% | 160,000 | 1.21 | 90.0% |
| C&W (white-box) | - | 0.88 | 99.0% | - | 0.25 | 85.0% | - | 1.51 | 80.0% |

of directly taking the sign of gradient estimation, our algorithm utilizes the scale of random direction $u$ as well. In other words, signSGD's gradient norm is always 1 while our gradient norm takes into account the magnitude of $u$. Therefore, our signOPT optimization algorithm is fundamentally different (Liu et al., 2019) or any other proposed signSGD varieties. Our method can be viewed as a new zeroth order optimization algorithm that features fast convergence in signSGD.

## 5 CONCLUSION

We developed a new and ultra query-efficient algorithm for adversarial attack in the hard-label black-box setting. Using the same smooth reformulation in Cheng et al. (2019), we design a novel zeroth order oracle that can compute the sign of directional derivative of the attack objective using single query. Equipped with this single-query oracle, we design a new optimization algorithm that can dramatically reduce number of queries compared with Cheng et al. (2019). We prove the convergence of the proposed algorithm and show our new algorithm is overwhelmingly better than current hard-label black-box attacks.

## ACKNOWLEDGEMENT

This work is based upon work supported by the Department of Energy National Energy Technology Laboratory under Award Number DE-OE0000911 and by NSF under IIS1719097.

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

# A APPENDIX

## A.1 COMPARISON WITH HOPSKIPJUMPATTACK

There is a recent paper (Chen et al., 2019) that applied the zeroth-order sign oracle to improve Boundary attack, and also demonstrated significant improvement. The major differences to our algorithm are that we propose a new zeroth-order gradient descent algorithm, provide its algorithmic convergence guarantees, and aim to improve the query complexity of the attack formulation proposed in (Cheng et al., 2019). To be noted, HopSkipJumpAttack only provides the bias and variance analysis (Theorem 2 and 3) without convergence rate analysis.

Also, HopSkipJumpAttack uses one-point gradient estimate compared to the 2-point gradient estimate used by SignOPT. Therefore, although the estimation is unbiased, it has large variance, which achieves successful attack faster but generates a worse adversarial example with larger distortion than ours. For completeness, we also compare with this method (and mention the results) as follows.

Figure 7 shows a comparison of Sign-OPT and HopSkipJumpAttack for CIFAR-10 and MNIST datasets for the case of $L_2$ norm based attack. We find in our experiments that performance of both attacks is comparable in terms of queries consumed. In some cases, Sign-OPT converges to a better solution.

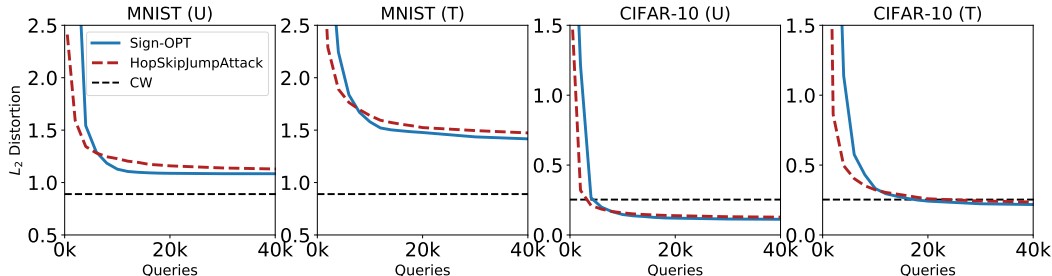

Figure 7: Comparison with HopSkipJumpAttack for CIFAR and MNIST: Median distortion vs Queries. (U) represents untargeted attack and (T) represents targeted attack.

## A.2 PROOF

Define following notations:

$$\hat{\nabla}g(\boldsymbol{\theta}_t; \boldsymbol{u}_q) \coloneqq \text{sign}(g(\boldsymbol{\theta}_t + \epsilon\boldsymbol{u}_q) - g(\boldsymbol{\theta}_t))\boldsymbol{u}_q$$

$$\dot{\nabla}g(\boldsymbol{\theta}_t; \boldsymbol{u}_q) \coloneqq \frac{1}{\epsilon}(g(\boldsymbol{\theta}_t + \epsilon\boldsymbol{u}_q) - g(\boldsymbol{\theta}_t))\boldsymbol{u}_q$$

$$\bar{\nabla}g(\boldsymbol{\theta}_t; \boldsymbol{u}_q) \coloneqq \text{sign}(\frac{1}{\epsilon}(g(\boldsymbol{\theta}_t + \epsilon\boldsymbol{u}_q) - g(\boldsymbol{\theta}_t))\boldsymbol{u}_q)$$

Thus we could write the corresponding estimate of gradients as follow:

$$\hat{\boldsymbol{g}}_t = \frac{1}{Q}\sum_{q=1}^{Q}\text{sign}(g(\boldsymbol{\theta}_t + \epsilon\mathbf{u}_q) - g(\boldsymbol{\theta}_t))\boldsymbol{u}_q = \frac{1}{Q}\sum_{q=1}^{Q}\hat{\nabla}g(\boldsymbol{\theta}_t; \boldsymbol{u}_q)$$

$$\dot{\boldsymbol{g}}_t = \frac{1}{Q}\sum_{q=1}^{Q}\frac{1}{\epsilon}(g(\boldsymbol{\theta}_t + \epsilon\boldsymbol{u}_q) - g(\boldsymbol{\theta}_t))\boldsymbol{u}_q = \frac{1}{Q}\sum_{q=1}^{Q}\dot{\nabla}g(\boldsymbol{\theta}_t; \boldsymbol{u}_q)$$

$$\bar{\boldsymbol{g}}_t = \frac{1}{Q}\sum_{q=1}^{Q}\text{sign}(\frac{1}{\epsilon}(g(\boldsymbol{\theta}_t + \epsilon\boldsymbol{u}_q) - g(\boldsymbol{\theta}_t))\boldsymbol{u}_q) = \frac{1}{Q}\sum_{q=1}^{Q}\bar{\nabla}g(\boldsymbol{\theta}_t; \boldsymbol{u}_q)$$

Clearly, we have $\bar{\nabla}g(\boldsymbol{\theta}_t; \boldsymbol{u}_q) = \text{sign}(\dot{\nabla}g(\boldsymbol{\theta}_t; \boldsymbol{u}_q))$ and we could relate $\bar{\nabla}g(\boldsymbol{\theta}_t; \boldsymbol{u}_q)$ and $\hat{\nabla}g(\boldsymbol{\theta}_t; \boldsymbol{u}_q)$ by writing $\hat{\nabla}g(\boldsymbol{\theta}_t; \boldsymbol{u}_q) = G_q \odot \bar{\nabla}g(\boldsymbol{\theta}_t; \boldsymbol{u}_q)$ where where $G_q \in \mathbb{R}^d$ is absolute value of vector $\boldsymbol{u}_q$ (i.e. $G_q = (|\boldsymbol{u}_{q,1}|, |\boldsymbol{u}_{q,2}|, \cdots, |\boldsymbol{u}_{q,d}|)^T)$.

Note that Zeroth-order gradient estimate $\dot{\nabla}g(\boldsymbol{\theta}_t; \boldsymbol{u}_q)$ is a biased approximation to the true gradient of g. Instead, it becomes unbiased to the gradient of the randomized smoothing function $g_\epsilon(\boldsymbol{\theta}) = \mathbb{E}_{\boldsymbol{u}}[g(\boldsymbol{\theta} + \epsilon \boldsymbol{u})]$ Duchi et al. (2012).

Our analysis is based on the following two assumptions:

**Assumption 1**   function g is L-smooth with a finite value of L.

**Assumption 2**   At any iteration step t, the gradient of the function g is upper bounded by $\|\nabla g(\boldsymbol{\theta}_t)\|_2 \leq \sigma$.

To prove the convergence of proposed method, we need the information on variance of the update $\dot{\nabla}g(\boldsymbol{\theta}_t; \boldsymbol{u}_q)$. Here, we introduce a lemma from previous works.

**Lemma 1**   The variance of Zeroth-Order gradient estimate $\dot{\nabla}g(\boldsymbol{\theta}_t; \boldsymbol{u}_q)$ is upper bounded by

$$\mathbb{E}\big[\|\dot{\nabla}g(\boldsymbol{\theta}_t; \boldsymbol{u}_q) - \nabla g_\epsilon(\boldsymbol{\theta}_t)\|_2^2\big] \leq \frac{4(Q+1)}{Q}\sigma^2 + \frac{2}{Q}C(d, \epsilon),$$

where $C(d, \epsilon) := 2d\sigma^2 + \epsilon^2 L^2 d^2/2$

**Proof of Lemma 1**   This lemma could be proved by using proposition 2 in Liu et al. (2019) with b = 1 and q = Q. When b = 1 there is no difference between with/without replacement, and we opt for with replacement case to obtain above bound.   □

By talking Q = 1, we know that $\mathbb{E}\big[\|\dot{\nabla}g(\boldsymbol{\theta}_t; \boldsymbol{u}_q) - \nabla g_\epsilon(\boldsymbol{\theta}_t)\|_2^2\big]$ is upper bounded. And by Jensen's inequality, we also know that the

$$\mathbb{E}\big[|(\dot{\nabla}g(\boldsymbol{\theta}_t; \boldsymbol{u}_q) - \nabla g_\epsilon(\boldsymbol{\theta}_t))_l|\big] \leq \sqrt{\mathbb{E}\big[((\dot{\nabla}g(\boldsymbol{\theta}_t; \boldsymbol{u}_q) - \nabla g_\epsilon(\boldsymbol{\theta}_t)))_l^2\big]} := \delta_l, \tag{6}$$

where $\delta_l$ denotes the upper bound of $lth$ coordinate of $\mathbb{E}\big[|\dot{\nabla}g(\boldsymbol{\theta}_t; \boldsymbol{u}_q) - \nabla g_\epsilon(\boldsymbol{\theta}_t)|\big]$, and $\delta_l$ is finite since $\mathbb{E}\big[\|\dot{\nabla}g(\boldsymbol{\theta}_t; \boldsymbol{u}_q) - \nabla g_\epsilon(\boldsymbol{\theta}_t)\|_2^2\big]$ is upper bounded.

Next, we want to show the $\text{Prob}[\text{sign}((\bar{\boldsymbol{g}}_t)_l) \neq \text{sign}((\nabla g_\epsilon(\boldsymbol{\theta}_t))_l)]$ by following lemma.

**Lemma 2**   $|(\nabla g_\epsilon(\boldsymbol{\theta}_t))_l|\text{Prob}[\text{sign}((\bar{\boldsymbol{g}}_t)_l) \neq \text{sign}((\nabla g_\epsilon(\boldsymbol{\theta}_t))_l)] \leq \frac{\delta_l}{\sqrt{Q}}$

**Proof of Lemma 2**   Similar to Bernstein et al. (2018), we first relax $\text{Prob}[\text{sign}((\dot{\nabla}g(\boldsymbol{\theta}_t; \boldsymbol{u}_q))_l) \neq \text{sign}(\nabla g_\epsilon(\boldsymbol{\theta}_t))_l]$ by Markov inequality:

$$\text{Prob}[\text{sign}((\dot{\nabla}g(\boldsymbol{\theta}_t; \boldsymbol{u}_q))_l) \neq \text{sign}((\nabla g_\epsilon(\boldsymbol{\theta}_t))_l)] \leq \text{Prob}[|\dot{\nabla}g(\boldsymbol{\theta}_t; \boldsymbol{u}_q)_l| \geq |\nabla g_\epsilon(\boldsymbol{\theta}_t)_l|]$$

$$\leq \frac{\mathbb{E}\big[|(\dot{\nabla}g(\boldsymbol{\theta}_t; \boldsymbol{u}_q) - \nabla g_\epsilon(\boldsymbol{\theta}_t))_l|\big]}{|\nabla g_\epsilon(\boldsymbol{\theta}_t)_l|}$$

$$\leq \frac{\delta_l}{|\nabla g_\epsilon(\boldsymbol{\theta}_t)_l|},$$

where the last inequality comes from eq (6). Recall that $(\dot{\nabla}g(\boldsymbol{\theta}_t; \boldsymbol{u}_q))_l$ is an unbiased estimation to $(\nabla g_\epsilon(\boldsymbol{\theta}_t))_l$. Under the assumption that the noise distribution is unimodal and symmetric, from Bernstein et al. (2018) Lemma D1, we will have

$$\text{Prob}[\text{sign}((\dot{\nabla}g(\boldsymbol{\theta}_t; \boldsymbol{u}_q))_l) \neq \text{sign}(\nabla g_\epsilon(\boldsymbol{\theta}_t))_l] := M \leq \begin{cases} \frac{2}{9}\frac{1}{S^2}, & S \geq \frac{2}{\sqrt{3}} \\ \frac{1}{2} - \frac{S}{2\sqrt{3}}, & otherwise \end{cases} < \frac{1}{2},$$

where $S := |\nabla g_\epsilon(\boldsymbol{\theta}_t)_l|/\delta_l$.

Note that this probability bound applies uniformly to all $q \in Q$ regardless of the magnitude $|(\boldsymbol{u}_q)_l|$. That is,

$$\text{Prob}[\text{sign}(\sum_{q=1}^{Q} |(\boldsymbol{u}_q)_l| \text{sign}((\dot{\nabla} g(\boldsymbol{\theta}_t; \boldsymbol{u}_q))_l) \neq \text{sign}(\nabla g_\epsilon(\boldsymbol{\theta}_t))_l] =$$

$$\text{Prob}[\text{sign}((\sum_{q=1}^{Q} \text{sign}(\dot{\nabla} g(\boldsymbol{\theta}_t; \boldsymbol{u}_q))_l) \neq \text{sign}(\nabla g_\epsilon(\boldsymbol{\theta}_t))_l]. \tag{7}$$

This is true as when all $|(\boldsymbol{u}_q)_l| = 1$, $\text{Prob}[\text{sign}((\sum_{q=1}^{Q} \text{sign}(\dot{\nabla} g(\boldsymbol{\theta}_t; \boldsymbol{u}_q))_l) \neq \text{sign}(\nabla g_\epsilon(\boldsymbol{\theta}_t))_l]$ is equivalent to majority voting of each estimate q yielding correct sign. This is the same as sum of Q bernoulli trials (i.e. binomial distribution) with error rate M. And since error probability M is independent of sampling of $|(\boldsymbol{u}_q)_l|$, calculating $\text{Prob}[\text{sign}(\sum_{q=1}^{Q} |(\boldsymbol{u}_q)_l| \text{sign}((\dot{\nabla} g(\boldsymbol{\theta}_t; \boldsymbol{u}_q))_l) \neq \text{sign}(\nabla g_\epsilon(\boldsymbol{\theta}_t))_l]$ could be thought as taking Q bernoulli experiments and then independently draw a weight from unit length for each of Q experiment. Since the weight is uniform, we will have expectation of weights on correct counts and incorrect counts are the same and equal to 1/2. Therefore, the probability of $\text{Prob}[\text{sign}(\sum_{q=1}^{Q} |(\boldsymbol{u}_q)_l| \text{sign}((\dot{\nabla} g(\boldsymbol{\theta}_t; \boldsymbol{u}_q))_l) \neq \text{sign}(\nabla g_\epsilon(\boldsymbol{\theta}_t))_l]$ is still the same as original non-weighted binomial distribution. Notice that by our notation, we will have $\text{sign}(\dot{\nabla} g(\boldsymbol{\theta}_t; \boldsymbol{u}_q)_l) = \bar{\nabla} g(\boldsymbol{\theta}_t; \boldsymbol{u}_q)_l$ thus $\frac{1}{Q} \sum_{q=1}^{Q} \text{sign}(\dot{\nabla} g(\boldsymbol{\theta}_t; \boldsymbol{u}_q))_l = (\bar{\boldsymbol{g}}_t)_l$. Let $Z$ counts the number of estimates $\dot{\nabla} g(\boldsymbol{\theta}_t; \boldsymbol{u}_q)_l$ yielding correct sign of $\nabla g_\epsilon(\boldsymbol{\theta}_t)_l$. Probability in eq (7) could be written as:

$$\text{Prob}[\text{sign}(\text{sign}((\bar{\boldsymbol{g}}_t)_l) \neq \text{sign}(\nabla g_\epsilon(\boldsymbol{\theta}_t))_l] = P[Z \leq \frac{Q}{2}].$$

Following the derivation of theorem 2b in Bernstein et al. (2018), we could get

$$P[Z \leq \frac{Q}{2}] \leq \frac{1}{\sqrt{Q}S}$$

$$\Rightarrow |(\nabla g_\epsilon(\boldsymbol{\theta}_t))_l| \text{Prob}[\text{sign}((\bar{\boldsymbol{g}}_t)_l) \neq \text{sign}((\nabla g_\epsilon(\boldsymbol{\theta}_t))_l)] \leq \frac{\delta_l}{\sqrt{Q}} \tag{8}$$

$\square$

We also need few more lemmas on properties of function g.

**Lemma 3**   $g_\epsilon(\boldsymbol{\theta}_1) - g_\epsilon(\boldsymbol{\theta}_T) \leq g_\epsilon(\boldsymbol{\theta}_1) - g^* + \epsilon^2 L$

**Proof of Lemma 3**   The proof can be found in Liu et al. (2018) Lemma C.   $\square$

**Lemma 4**   $\mathbb{E}[\|\nabla g(\boldsymbol{\theta})\|_2] \leq \sqrt{2}\mathbb{E}[\|\nabla g_\epsilon(\boldsymbol{\theta})\|_2] + \frac{\epsilon L d}{\sqrt{2}}$, where $g^* = \min_{\boldsymbol{\theta}} g(\boldsymbol{\theta})$.

**Proof of Lemma 4**   The proof can be found in Liu et al. (2019).   $\square$

**Theorem 1**   Suppose that the conditions in the assumptions hold, and the distribution of gradient noise is unimodal and symmetric. Then, Sign-OPT attack with learning rate $\eta_t = O(\frac{1}{Q\sqrt{dT}})$ and $\epsilon = O(\frac{1}{dT})$ will give following bound on $\mathbb{E}[\|\nabla g(\boldsymbol{\theta})\|_2]$

$$\mathbb{E}[\|\nabla g(\boldsymbol{\theta})\|_2] = O(\frac{\sqrt{d}}{\sqrt{T}} + \frac{d}{\sqrt{Q}})$$

**Proof of Theorem 1**   From L-smoothness assumption we could have

$$g_\epsilon(\boldsymbol{\theta}_{t+1}) \le g_\epsilon(\boldsymbol{\theta}_t) + \langle \nabla g_\epsilon(\boldsymbol{\theta}_t), \boldsymbol{\theta}_{t+1} - \boldsymbol{\theta}_t \rangle + \frac{L}{2} \|\boldsymbol{\theta}_{t+1} - \boldsymbol{\theta}_t\|_2^2$$

$$= g_\epsilon(\boldsymbol{\theta}_t) - \eta_k \langle \nabla g_\epsilon(\boldsymbol{\theta}_t), \hat{\boldsymbol{g}}_t \rangle + \frac{L}{2} \eta_t^2 \|\hat{\boldsymbol{g}}_t\|_2^2$$

$$= g_\epsilon(\boldsymbol{\theta}_t) - \eta_t \odot \bar{G}_t \|\nabla g_\epsilon(\boldsymbol{\theta}_t)\|_1 + \frac{dL}{2} \eta_t^2 \odot \bar{G}_t^2$$

$$+ 2\eta_t \odot \bar{G}_t \sum_{l=1}^d |(\nabla g_\epsilon(\boldsymbol{\theta}_t))_l| \text{Prob}[\text{sign}((\bar{\boldsymbol{g}}_t)_l) \ne \text{sign}((\nabla g_\epsilon(\boldsymbol{\theta}_t))_l)],$$

where $\bar{G}_t$ is defined as $(\bar{G}_t)_l = \sum_{q=1}^Q (G_q)_l \bar{\nabla} g(\boldsymbol{\theta}_t; \boldsymbol{u}_q)_l = \sum_{q=1}^Q |(\boldsymbol{u}_q)_l| \bar{\nabla} g(\boldsymbol{\theta}_t; \boldsymbol{u}_q)_l$. Continue the inequality,

$$g_\epsilon(\boldsymbol{\theta}_t) - \eta_t \odot \bar{G}_t \|\nabla g_\epsilon(\boldsymbol{\theta}_t)\|_1 + \frac{dL}{2} \eta_t^2 \odot \bar{G}_t^2$$

$$+ 2\eta_t \odot \bar{G}_t \sum_{l=1}^d |(\nabla g_\epsilon(\boldsymbol{\theta}_t))_l| \text{Prob}[\text{sign}((\bar{\boldsymbol{g}}_t)_l) \ne \text{sign}((\nabla g_\epsilon(\boldsymbol{\theta}_t))_l)]$$

$$\le g_\epsilon(\boldsymbol{\theta}_t) - \eta_t \odot \bar{G}_t \|\nabla g_\epsilon(\boldsymbol{\theta}_t)\|_1 + \frac{dL}{2} \eta_t^2 \odot \bar{G}_t^2 + 2\eta_t \odot \bar{G}_t \sum_{l=1}^d \frac{\delta_l}{\sqrt{Q}} \qquad \text{by eq (8)}$$

$$\le g_\epsilon(\boldsymbol{\theta}_t) - \eta_t \odot \bar{G}_t \|\nabla g_\epsilon(\boldsymbol{\theta}_t)\|_1 + \frac{dL}{2} \eta_t^2 \odot \bar{G}_t^2 + 2\eta_t \odot \bar{G}_t \frac{\|\delta_l\|_1}{\sqrt{Q}}$$

$$\le g_\epsilon(\boldsymbol{\theta}_t) - \eta_t \odot \bar{G}_t \|\nabla g_\epsilon(\boldsymbol{\theta}_t)\|_1 + \frac{dL}{2} \eta_t^2 \odot \bar{G}_t^2 + 2\eta_t \odot \bar{G}_t \frac{\sqrt{d}\sqrt{\|\delta_l\|_2^2}}{\sqrt{Q}}$$

$$= g_\epsilon(\boldsymbol{\theta}_t) - \eta_t \odot \bar{G}_t \|\nabla g_\epsilon(\boldsymbol{\theta}_t)\|_1 + \frac{dL}{2} \eta_t^2 \odot \bar{G}_t^2 + 2\eta_t \odot \bar{G}_t \frac{\sqrt{d}\sqrt{\mathbb{E}[((\dot{\nabla} g(\boldsymbol{\theta}_t; \boldsymbol{u}_q) - \nabla g_\epsilon(\boldsymbol{\theta}_t)))_l^2]}}{\sqrt{Q}} \quad \text{by eq (6).}$$

Thus we will have,

$$g_\epsilon(\boldsymbol{\theta}_{t+1}) - g_\epsilon(\boldsymbol{\theta}_t) \le -\eta_t \odot \bar{G}_t \|\nabla g_\epsilon(\boldsymbol{\theta}_t)\|_1 + \frac{dL}{2} \eta_t^2 \odot \bar{G}_t^2 + 2\eta_t \odot \bar{G}_t \frac{\sqrt{d}\sqrt{\mathbb{E}[((\dot{\nabla} g(\boldsymbol{\theta}_t; \boldsymbol{u}_q) - \nabla g_\epsilon(\boldsymbol{\theta}_t)))_l^2]}}{\sqrt{Q}}$$

$$\Rightarrow \eta_t \odot \bar{G}_t \|\nabla g_\epsilon(\boldsymbol{\theta}_t)\|_1 \le g_\epsilon(\boldsymbol{\theta}_t) - g_\epsilon(\boldsymbol{\theta}_{t+1}) + \frac{dL}{2} \eta_t^2 \odot \bar{G}_t^2 + 2\eta_t \odot \bar{G}_t \frac{\sqrt{d}\sqrt{\mathbb{E}[((\dot{\nabla} g(\boldsymbol{\theta}_t; \boldsymbol{u}_q) - \nabla g_\epsilon(\boldsymbol{\theta}_t)))_l^2]}}{\sqrt{Q}}$$

$$\Rightarrow \hat{\eta}_t \|\nabla g_\epsilon(\boldsymbol{\theta}_t)\|_1 \le g_\epsilon(\boldsymbol{\theta}_t) - g_\epsilon(\boldsymbol{\theta}_{t+1}) + \frac{dL}{2} \hat{\eta}_t^2 + 2\hat{\eta}_t \sqrt{d} \frac{\sqrt{\mathbb{E}[((\dot{\nabla} g(\boldsymbol{\theta}_t; \boldsymbol{u}_q) - \nabla g_\epsilon(\boldsymbol{\theta}_t)))_l^2]}}{\sqrt{Q}},$$

where we define $\hat{\eta}_t := \eta_t \odot \bar{G}_t$. Sum up all inequalities for all ts and take expectation on both side, we will have

$$\sum_{t=1}^T \hat{\eta}_t \mathbb{E}[\|\nabla g_\epsilon(\boldsymbol{\theta}_t)\|_1] \le \mathbb{E}[g_\epsilon(\boldsymbol{\theta}_1) - g_\epsilon(\boldsymbol{\theta}_T)] + \frac{dL}{2} \sum_{t=1}^T \hat{\eta}_t^2 + \sum_{t=1}^T 2\hat{\eta}_t \sqrt{d} \sqrt{\mathbb{E}[((\dot{\nabla} g(\boldsymbol{\theta}_t; \boldsymbol{u}_q) - \nabla g_\epsilon(\boldsymbol{\theta}_t)))_l^2]}$$

$$\le \mathbb{E}[g_\epsilon(\boldsymbol{\theta}_1) - g_\epsilon(\boldsymbol{\theta}_T)] + \frac{dL}{2} \sum_{t=1}^T \hat{\eta}_t^2 + \sum_{t=1}^T 2\hat{\eta}_t \sqrt{d} \sqrt{\frac{4(Q+1)}{Q} \sigma^2 + \frac{2}{Q} C(d, \epsilon)} \qquad \text{by Lemma 1.}$$

Substitute Lemma 3 into above inequality, we get

$$\sum_{t=1}^T \hat{\eta}_t \mathbb{E}[\|\nabla g_\epsilon(\boldsymbol{\theta}_t)\|_1] \le g_\epsilon(\boldsymbol{\theta}_1) - g^* + \epsilon^2 L + \frac{dL}{2} \sum_{t=1}^T \hat{\eta}_t^2 + \sum_{t=1}^T 2\hat{\eta}_t \sqrt{d} \sqrt{\frac{4(Q+1)}{Q} \sigma^2 + \frac{2}{Q} C(d, \epsilon)}.$$

Since $\| \cdot \|_2 \le \| \cdot \|_1$ and we could divide $\sum_{t=1}^{T} \hat{\eta}_t$ on both side to get

$$\sum_{t=1}^{T} \frac{\hat{\eta}_t}{\sum_{t=1}^{T} \hat{\eta}_t} \mathbb{E}[\|\nabla g_\epsilon(\boldsymbol{\theta}_t)\|_2] \le \frac{g_\epsilon(\boldsymbol{\theta}_1) - g^* + \epsilon^2 L}{\sum_{t=1}^{T} \hat{\eta}_t} + \frac{dL}{2} \frac{\sum_{t=1}^{T} \hat{\eta}_t^2}{\sum_{t=1}^{T} \hat{\eta}_t} + \sum_{t=1}^{T} \frac{2\sqrt{d}}{\sqrt{Q}} \sqrt{4(Q+1)\sigma^2 + 2C(d,\epsilon)}.$$

Define a new random variable R with probability $P(R = t) = \frac{\eta_t}{\sum_{t=1}^{T} \eta_t}$, we will have

$$\mathbb{E}[\|\nabla g_\epsilon(\boldsymbol{\theta}_R)\|_2] = \mathbb{E}[\mathbb{E}_R[\|\nabla g_\epsilon(\boldsymbol{\theta}_R)\|_2]] = \mathbb{E}\Big[ \sum_{t=1}^{T} P(R = t) \|\nabla g_\epsilon(\boldsymbol{\theta}_t)\|_2 \Big].$$

Substitute all the quantities into Lemma 4, we will get

$$\mathbb{E}[\|\nabla g(\boldsymbol{\theta})\|_2] \le \frac{\sqrt{2}(g_\epsilon(\boldsymbol{\theta}_1) - g^* + \epsilon^2 L)}{\sum_{t=1}^{T} \hat{\eta}_t} + \frac{dL}{\sqrt{2}} \frac{\sum_{t=1}^{T} \hat{\eta}_t^2}{\sum_{t=1}^{T} \hat{\eta}_t} + \frac{\epsilon Ld}{\sqrt{2}} + \sum_{t=1}^{T} \frac{2\sqrt{2}\sqrt{d}}{\sqrt{Q}} \sqrt{4(Q+1)\sigma^2 + 2C(d,\epsilon)}.$$

By choosing $\epsilon = O(\frac{1}{dT})$ and $\eta_t = O(\frac{1}{Q\sqrt{dT}})$, then the convergence rate as shown in above is $O(\frac{d}{T} + \frac{d}{\sqrt{Q}})$. $\quad \square$