# OpenReview forum: "Sign-OPT: A Query-Efficient Hard-label Adversarial Attack"
_ICLR.cc/2020/Conference — Accept (Poster)_

### Official Review · AnonReviewer1 · 2019-10-16
**Official Blind Review #1054**

**Rating:** 6

**Review:**

In this paper, the authors propose a new algorithm for evaluating adversarial robustness of black-box models. The aim of the proposed algorithm, SIGN-OPT is generating adversarial examples as close as possible to the decision boundary using as less queries of the black-box model as possible.

The authors follow the approach of (Cheng et al 2019) by modeling the problem as an optimization problem, where the objective function is to find the direction with the shortest distance to the decision boundary. They propose a smart modification of the previous approach by evaluating the sign of the gradient rather than the gradient itself. The advantage is that the sign of the gradient can be evaluated using a single query while the estimation of the gradient needs many queries.
The authors have analyzed the proposed algorithm. They showed that using SIGN-OPT, the expectation of the gradient tends to zero in $O(1/\sqrt(T))$, meaning that a (local) minimum is reached.
The algorithm is favorably compared with the state-of-the-art on three image test sets (MNIST, CIFAR-10n ImageNet).

This paper is technically sound, well-written and propose an interesting modification of a previous algorithm. I vote for acceptance.

However, I have some concerns:
-	I think that the L-smoothness assumption should be discussed. Is it realistic for Deep Learning models? Does it hold for the three attacked CNN networks?
-	The analytical results of the previous algorithm RGF and the proposed algorithm SIGN-OPT are not compared and discussed. It is a pity.



**Experience Assessment:**

I have read many papers in this area.

**Review Assessment: Checking Correctness Of Derivations And Theory:**

I carefully checked the derivations and theory.

**Review Assessment: Checking Correctness Of Experiments:**

I assessed the sensibility of the experiments.

**Review Assessment: Thoroughness In Paper Reading:**

I read the paper thoroughly.

---

> ### Author Response · Authors · 2019-11-11
> **Response to Reviewer #1**
>
> We thank the reviewer for valuable comments and suggestions.
>
> About the L-smoothness: This assumption is common for convergence analysis of both first-order and zeroth-order methods for nonconvex optimization, e.g., [3,4] since it is a common starting point to bound the stationary gap. And we couldn’t prove g(theta) is continuous for general deep neural networks. It’s possible that the g(theta) may not be continuous; for example, we think it might be possible to construct some counter-examples using ReLU activation. However, although the assumption may not hold for DNN globally, our algorithm still performs well in practice. What we can assure is that if $g(\cdot)$ has a Lipschitz continuous gradient, our algorithm has such a theoretical guarantee. This is indeed a sufficient but not necessary condition.
>
>
>
> About the analytical results:
> For conventional RGF methods such as ZO-SGD [1] and ZO-stochastic mirror descent [2], the convergence error is typically given by E[|\nabla f(x)|_2^2]=O(\sqrt{d}/\sqrt{T}). For ZO based sign-SGD, it was shown in [4] that the rate O(\sqrt{d}/\sqrt{T}) is actually improved since the error is bounded by the non-squared gradient norm E[|\nabla f(x)|_2] = O(\sqrt{d}/\sqrt{T} + d/\sqrt{q}) at the cost that sign-based ZO method converges to a neighborhood of a solution, whose size is determined by d and q. The work [4] showed that the query efficiency of generate black-box attacks seems a more important metric than the accuracy of the attack, since one cares more about whether or not the attack can succeed as early as possible. Compared to ZO-SignSGD, our rate yields similar pros and cons in Big O notations. However, we observe that it works much better than ZO-SignSGD in all of our experiments. More importantly, ZO-SignSGD was not designed for hard-label black-box attack, and used a more involved query oracle.  Furthermore, if we apply single query oracle to improve Liu's method,  since Liu's method takes sign for each element of the query direction, then it cannot fully utilize the single query oracle and leads to suboptimal convergence. To show this, in Figure 5 (b) we compare with Liu+Single-query-oracle and show Sign-OPT converges faster and achieves a better solution. We have added this comparison in the revision.
>
>
> [1]S. Ghadimi and G. Lan, “Stochastic first-and zeroth-order methods for nonconvex stochastic
> programming,” SIAM Journal on Optimization, vol. 23, no. 4, pp. 2341–2368, 2013
> [2]J. C. Duchi, M. I. Jordan, M. J. Wainwright, and A. Wibisono, “Optimal rates for zero-order convex optimization: The power of two function evaluations,”
> [3]Bernstein, Jeremy, et al. "signSGD: Compressed optimisation for non-convex problems." arXiv preprint arXiv:1802.04434 (2018).
> [4]Liu, Sijia, et al. "signSGD via zeroth-order oracle." (2018).

---

### Official Review · AnonReviewer3 · 2019-10-23
**Official Blind Review #3**

**Rating:** 3

**Review:**

Summary: The paper develops a query efficient algorithm for computing black box adversarial examples given only hard labels in the context of deep neural networks. Intuitively, the only information of the function provided to the algorithm is the label for a given sample. Technically, the authors use the formulation proposed by Cheng et al, and derive a zeroth optimization algorithm that uses less queries with nice convergence properties. Experimentally, the proposed algorithm is very effective on three different standard datasets in vision.

I have decided to weak reject the paper for the following key reasons:

1. Novelty: the technique as such as very similar to Cheng et al, and Liu et al, as the authors themselves mention it in Section 3.2. In particular, the speed-up compared to Cheng et al, is twice -- for a bounded maximum \alpha in Algorithm 1 in Cheng et al, which is almost always the case, because otherwise it would not be "adversarial" in nature. The authors claim that the convergence result has not been proved yet for the proposed algorithm but it follows using the technique used in Bernstein et al 2018, with some minor modifications.

2. Experiments: While the experiments that the authors support the claim, I think they are missing comparison with Liu et al which is crucial. In Fig 4, their method is doing even better than white box attacks, how can this be true? or why is this true?

**Experience Assessment:**

I have published one or two papers in this area.

**Review Assessment: Checking Correctness Of Derivations And Theory:**

I carefully checked the derivations and theory.

**Review Assessment: Checking Correctness Of Experiments:**

I carefully checked the experiments.

**Review Assessment: Thoroughness In Paper Reading:**

I read the paper thoroughly.

---

> ### Author Response · Authors · 2019-11-11
> **Response to Reviewer #3**
>
> We thank the reviewer for valuable comments and suggestions.
>
> About the novelty: Although we use the similar loss formulation as Cheng et al, the major difference is the proposal of single query oracle (section 3.1) that substantially reduces the cost of gradient sign estimation, and in section 3.2 and 3.3 we showed that this single query oracle can be used to solve Cheng’s objective function. We believe both of these are novel. In contrast, Cheng’s method requires binary search to estimate the loss and gradient thereby increasing the queries required.
>
> Regarding comparisons to Liu et al, we note that their algorithm is proposed for soft label black box setting, not for hard-label setting. More specifically, in their paper the black-box query will return *probability output* of the model, which is different from the hard-label query case studied in this paper. So the query counts are not comparable.
> It’s non-trivial to extend a soft-label black box attack to the hard-label setting since they are using totally different information. This is why there are tens of soft-label black box attack methods but only few hard-label attack methods. Again, our novelty is in proposing the single query oracle and showing that using such oracle, a query-efficient gradient-based update can be derived to solve Cheng’s hard-label attack formulation. The relationship between our paper and (Liu et al) is that our convergence analysis is a combination of (Liu et al) with a newly introduced way to handle magnitudes of each element of u_k.
>
> Also, we have added a discussion in Section 4.4 in the revision to discuss the importance of single query oracle and the improvement from sign-OPT over (Liu et al) with single query oracle.
>
>
>
> About the speed-up compared to Cheng et al:
> Sorry for the confusion. For OPT-attack, to evaluate g(\theta+\beta u) in each gradient estimation, even with the optimal alpha, we still need to do several binary search to achieve v_right-v_left<\epsilon, which would take extra number of queries. On the other hand, sign-OPT only uses single query. Moreover, in the implementation of both OPT-attack and Sign-OPT, instead of using one vector, they sample 10-20 vectors from Gaussian distribution and average their estimators. Therefore, Sign-OPT further reduces the number of queries several times. The experiments also show Sign-OPT achieve 5-10 times fewer queries in practice.
>
>
>
> About the white-box performance:
> Our method can sometimes outperform white-box attack in terms of distortion, The same finding has also been reported in Boundary attack (Brendel et al., 2018), see the table in their section 3.1. Several recent papers (including other submissions to ICLR this year) also reported similar findings and try to explain this phenomenon. We think the main reason is that hard-label attacks are based on searching on the decision surface, while C&W/PGD attacks are starting from the interior and gradually moving to the boundary, so they have very different optimization paths which lead to different local minima.

---

### Public Comment · ~Xingjun_Ma1 · 2020-03-08
**Sign black-box attack**

Nice work! Looks related to our probability black-box attack: "Black-box Adversarial Attacks on Video Recognition Models" (https://arxiv.org/abs/1904.05181), where we use NES to estimate the sign of the gradients for efficient black-box attack. But our estimation is based on the probabilities. Would be nice to see some discussions.

---

### Decision · Program_Chairs · 2019-12-19

**Decision:**

Accept (Poster)

**Comment:**

The reviewers had several concerns with the paper related to novelty and comparisons with other approaches. During the discussion phase, these concerns were adequately addressed.